# Vestibular syndromes, diagnosis and diagnostic errors in patients with dizziness presenting to the emergency department: a cross-sectional study

Lukas Comolli,[1] Athanasia Korda,[1] Ewa Zamaro,[1] Franca Wagner,[2] Thomas C Sauter,[3] Marco D Caversaccio,[1] Florence Nikles,[1] Simon Jung,[4] Georgios Mantokoudis  [1]

For numbered affiliations see end of article.

**Correspondence to**
Professor Dr Georgios Mantokoudis;
Georgios.Mantokoudis@insel.ch

## ABSTRACT

**Objectives** We aimed to determine the frequency of vestibular syndromes, diagnoses, diagnostic errors and resources used in patients with dizziness in the emergency department (ED).

**Design** Retrospective cross-sectional study.

**Setting** Tertiary referral hospital.

**Participants** Adult patients presenting with dizziness.

**Primary and secondary outcome measures** We collected clinical data from the initial ED report from July 2015 to August 2020 and compared them with the follow-up report if available. We calculated the prevalence of vestibular syndromes and stroke prevalence in patients with dizziness. Vestibular syndromes are differentiated in acute (AVS) (eg, stroke, vestibular neuritis), episodic (EVS) (eg, benign paroxysmal positional vertigo, transient ischaemic attack) and chronic (CVS) (eg, persistent postural-perceptual dizziness) vestibular syndrome. We reported the rate of diagnostic errors using the follow-up diagnosis as the reference standard.

**Results** We included 1535 patients with dizziness. 19.7% (303) of the patients presented with AVS, 34.7% (533) with EVS, 4.6% (71) with CVS and 40.9% (628) with no or unclassifiable vestibular syndrome. The three most frequent diagnoses were stroke/minor stroke (10.1%, 155), benign paroxysmal positional vertigo (9.8%, 150) and vestibular neuritis (9.6%, 148). Among patients with AVS, 25.4% (77) had stroke. The cause of the dizziness remained unknown in 45.0% (692) and 18.0% received a false diagnosis. There was a follow-up in 662 cases (43.1%) and 58.2% with an initially unknown diagnoses received a final diagnosis. Overall, 69.9% of all 1535 patients with dizziness received neuroimaging (MRI 58.2%, CT 11.6%) in the ED.

**Conclusions** One-fourth of patients with dizziness in the ED presented with AVS with a high prevalence (10%) of vestibular strokes. EVS was more frequent; however, the rate of undiagnosed patients with dizziness and the number of patients receiving neuroimaging were high. Almost half of them still remained without diagnosis and among those diagnosed were often misclassified. Many unclear cases of vertigo could be diagnostically clarified after a follow-up visit.

## STRENGTHS AND LIMITATIONS OF THIS STUDY

⇒ This cross-sectional study includes a large number of patients with dizziness visiting the emergency department.

⇒ We report the frequency of vestibular syndromes based on the international classification of the Bárány Society.

⇒ For a more accurate classification of vestibular syndromes, a prospective longitudinal study design would be needed.

⇒ We observed a referral bias (tertiary referral centre) leading to a higher proportion of dangerous diagnoses in patients with dizziness.

⇒ Since the treating clinician decided whether a follow-up was pursued, there might be a selection bias.

## BACKGROUND

Patients with dizziness presenting in the emergency department (ED) often suffer from accompanying symptoms such as nausea, vomiting, gait disturbance and motion intolerance, summarised as a vestibular syndrome.[1] There is no direct link to a specific cause such as a peripheral or central disorder;[2] however, physicians might narrow down their differential diagnosis by classifying it into three basic categories of vestibular syndromes:[3] episodic, acute and chronic. Such classification is based on the time course and duration of symptoms as well as on whether the symptoms are continuous or repetitive. This means a paradigm shift from classical teaching,[4] which is focusing on history taking and investigating symptom qualities such as vertigo, disequilibrium, presyncope and non-specific dizziness. Previous investigations proved that the description of symptom quality is imprecise and inaccurate for diagnostic decisions.[5] The classification of different vestibular syndromes is internationally accepted and

was introduced in the recently revised International Classification of Diseases (ICD) from the WHO (ICD-11 and ICD-12 codes, 2016).[6] This new definition was elaborated by the international and interdisciplinary Bárány Society. It allows physicians to recognise patterns, to apply different diagnostic tests based on their classification and to reduce the number of differential diagnoses; however, the frequency of vestibular syndromes and their underlying diagnosis remains poorly investigated. In addition, there is an expected overlap of timing and symptoms within each syndrome since any acute vestibular syndrome might persist and develop into a chronic disease or might occur repetitively with symptom-free intervals.

We therefore sought to investigate the frequency of vestibular syndromes to assess the underlying diagnosis stratified by syndromes, the frequency of diagnostic errors comparing the initial with the follow-up visit and to describe the resource consumption in the ED.

## METHODS

In this retrospective cross-sectional study, we used data collected prospectively during screening for the DETECT (Dizziness Evaluation Tool for Emergent Clinical Triage) study.[7–10] The sample size for this study was given through the DETECT study, where a sample size of 200 patients with an acute vestibular syndrome (AVS) was needed. We used the screening data which were needed to recruit these 200 patients. We were looking for patients presenting to the ED of the Inselspital Bern (University Hospital and tertiary referral hospital) with an AVS and a suspected stroke diagnosis. Research fellows trained in neuro-otology prospectively screened and identified patients with dizziness during daytime hours from July 2015 to August 2020 using either the ED triage software system (chief complaints such as 'dizziness', 'vertigo', 'unsteadiness', 'presyncope', 'vomiting', 'nausea' or a suspected diagnosis) or direct information from the emergency physician. We included all ED patients presenting with dizziness older than 16 years (ED index visit). We use dizziness as an umbrella term throughout the manuscript including the following set of symptoms: vertigo, dizziness, gait or balance unsteadiness, ataxia and syncope or presyncope. We collected data about baseline demographics, medical history, clinical findings, resources used and diagnoses. In a second step, we retrospectively compared data from the index visit in the ED with data collected from patients who received a follow-up examination at our hospital's dizziness clinic within 90 days after presentation to the ED (follow-up visit).

### Classification of vestibular syndromes

We classified all included patients into five categories based on the International Classification from the Bárány Society[1] and predefined criteria:[3] (1) acute, (2) episodic and (3) chronic vestibular syndrome, (4) acute imbalance syndrome and (5) patients not classifiable (unclear). We defined vestibular syndromes below.

### Acute vestibular syndrome

AVS is defined as a clinical syndrome of acute onset, continuous dizziness lasting days to weeks and generally including features suggestive of new, ongoing vestibular system dysfunction (eg, vomiting, nystagmus, severe postural instability).[1] Although this syndrome is characterised by a single, monophasic event due to a one-time disorder, it might be the beginning of a recurrent disease or a progressive illness course. Thus, AVS might overlap with other syndromes explained below or change over time. There are subclassifications of AVS mentioned in the literature[11] such as t-AVS (postexposure dizziness after trauma or toxic exposure) or s-AVS (spontaneous AVS) including all patients with continuous dizziness at rest. For the sake of simplicity, we classified all these patients under the umbrella term AVS.

### Episodic vestibular syndrome

The episodic vestibular syndrome (EVS) is characterised as transient dizziness lasting seconds to hours, rarely days. It is accompanied by a short duration of nausea, nystagmus and sudden falls.[1] EVS can occur repetitively (episodes) caused by an episodic disorder with repeated spells or as a single event (first manifestation) of a progressive chronic disorder with transient or recurrent dizziness. There are subtypes of EVS with associated triggers (t-EVS) or without triggers (s-EVS, spontaneous EVS). Diagnosis of s-EVS is mainly based on the patient's history. Patients with t-EVS have often clinical signs such as positional nystagmus after provocation. Both subgroups were included as EVS without separate differentiation.

### Chronic vestibular syndrome

The chronic vestibular syndrome (CVS) lasts usually months to years and is generally associated with a persistent vestibular system dysfunction (eg, oscillopsia, nystagmus, gait unsteadiness, falls).

### Acute imbalance/disbalance syndrome

Patients with symptoms that did not meet definitions 1–3 and therefore a vestibular syndrome could be excluded were classified as an acute imbalance syndrome (AIS).[12 13] Patients with dizziness as an isolated symptom and no accompanying symptoms or no nystagmus were, therefore, classified as 'AIS'.

### Unclear vestibular syndrome

If the information in the medical report was not specific enough to decide whether it was a vestibular syndrome or not, they were labelled as 'unclear'.

The type of syndromes and diagnoses from the index visit (ED diagnosis) and the follow-up examination (follow-up or final diagnosis) were analysed and compared, if available. We only included the main diagnosis reasonable for causing dizziness, and additional diagnoses were classified as 'other diagnoses'. Patients with more than one differential diagnosis causing dizziness were classified as 'unknown'. Patients were reclassified regarding the type of vestibular syndrome based on the time course of

symptoms and signs. Patients, for example, with symptoms lasting less than 24 hours or with repetitive events were reported or reclassified as EVS. Patients with misclassified EVS were often sent home within a few hours after symptom onset. Initially, misclassified EVS with persistent symptoms, however, were reclassified as AVS.

We calculated the overall rate of diagnostic errors between the initial ED diagnosis and the follow-up diagnosis using the follow-up diagnosis as the reference standard. We also reported the change of diagnoses rate stratified by ED diagnoses. The rate of changes of diagnoses at follow-up was calculated as follows: 100*(1−correct diagnoses/total diagnoses ED). The diagnosis was assumed to be correct if it did not change from the initial to the follow-up diagnosis.

## Statistics

We used SPSS (IBM Released 2020, IBM SPSS Statistics for Windows, V.27.0; IBM, Armonk, New York, USA) for statistics and descriptive data analysis. We did a subgroup analysis on those patients who received a follow-up examination. Cross-tabulations were used to compare results at the ED index visit with the follow-up visit. Cohen's kappa was calculated to report the concordance between index visit and follow-up regarding the classification of vestibular syndromes and diagnoses. We defined a change in the diagnosis at the follow-up as a diagnostic error.

## Patient and public involvement

Patients or the public were not involved in our research design, conduct, reporting or dissemination plans.

## RESULTS

### Prevalence of vestibular syndromes and underlying diagnoses

We included 1535 patients aged from 16 to 98 (mean 55.7 years±SD 18.6 years) who presented with dizziness as a chief complaint. Our cohort consisted of 745 (48.5%) men and 790 (51.5%) women. The age and gender distribution are shown as a histogram in the online supplemental figure S1. Of all patients, 303 presented with AVS (19.7%), 533 with EVS (34.7%), 71 with CVS (4.6%) and 472 patients had an AIS (30.8%). In 156 cases (10.2%), the type of vestibular syndrome remained unclear or was not classifiable based on clinical and reported findings. Since several diagnoses could be selected, there were more diagnoses than cases.

The five most frequent diagnoses including all types of vestibular syndromes were strokes (n=155, 10.1%), benign paroxysmal positional vertigo (BPPV) (n=150, 9.8%), acute unilateral vestibulopathy (n=148, 9.6%), transient ischaemic attack (TIA) (n=77, 5.0%) and dysautonomia (n=63, 4.1%). In 692 cases (45.0%) the diagnosis remained unknown. A dysautonomia was diagnosed when the 'Schellong test' was positive.[14] Table 1 shows the frequency of diagnoses stratified by vestibular syndromes.

### Accuracy of syndrome classification

Out of 1535 patients, 662 (43.1%) received a follow-up. There was an excellent agreement (Cohen's kappa=0.909,

p<0.001) between the syndrome classification at index visit and follow-up with a reported change of the AVS in 3.2% after the follow-up. Most of the patients with misclassified AVS were reassessed as EVS. Patients with EVS, however, were misclassified in 3.6%. Among the patients with an AIS on the ED, the reclassification rate was 8.0%, whereas one patient was subsequently classified as AVS. In the cases that could not be initially classified in the ED, 34.7% could be classified as a vestibular syndrome or AIS in the follow-up examination (table 2).

### Diagnostic errors in patients with dizziness

In this section, we compare the diagnosis at ED with the diagnosis at follow-up (n=662). We report an overall change in diagnosis between initial ED assessment and follow-up of 31.4%. The proportion of diagnostic errors (excluding patients with unknown causes) was 18.0%. There was a moderate to low agreement between the initial diagnosis (ED diagnosis) and the final diagnosis after the follow-up (Cohen's kappa=0.609, p<0.001). Often diagnostic errors occurred in patients with dysautonomia (33%, 6/9), TIA (30.6%, 15/49), BPPV (28.6%, 8/28), Menière's disease (26.7%, 4/15), stroke/minor stroke (13.6%, 18/132) and for acute unilateral vestibulopathy (15.7%, 14/89). Of the cases with an initial diagnosis of TIA, the diagnosis was changed during follow-up to 'stroke/minor stroke' in seven and to 'unknown' in four cases (table 3 and online supplemental table S1). The cause of the dizziness at the time of the ED visit was unknown in 37.6%. In 104 out of 662 cases, the diagnosis remained unclear even after the follow-up examination, however, 58.2% of all unknown cases in the ED received finally a diagnosis and could be clarified (table 4). A special focus was placed on patients with an undiagnosed dangerous cause of dizziness (strokes/minor strokes, TIA) leading to potential diagnosis-related harm. There were two patients initially diagnosed with BBPV, three with acute unilateral vestibulopathy and one case with a medical side effect where the initial diagnosis was changed to TIA or a stroke/minor stroke at follow-up. Among patients with no specific diagnoses in the ED (classified as unknown/unclear), 14 patients had a stroke and 9 had a TIA. In summary, in 29 of the 662 followed up cases (4.4%), a dangerous diagnosis was found at follow-up (potential diagnosis-related harms) which was initially not diagnosed in the ED (online supplemental table S1, bold cases).

### ED resource use

Overall, 69.9% of all 1535 patients with dizziness received neuroimaging at the ED visit (MRI 58.2%, CT 11.6%). 16.8% of patients with stroke underwent a CT and 89.7% had an MRI. Patients with BPPV received MRI in 41.3% and CT in 8%, showing a similar resource use as patients with acute unilateral vestibulopathy (48% MRI, 6.8% CT). Table 5 shows details of ED resource use stratified by ED diagnoses.

**Table 1** ED diagnoses stratified by vestibular syndromes

| Diagnosis | Total (N=1535) | AVS (n=303) | EVS (n=533) | CVS (n=71) | AIS (n=472) | Unclear (n=156) |
|---|---|---|---|---|---|---|
| Stroke/minor stroke | 155 (10.10%) | 77 (25.41%) | 10 (1.88%) | 2 (2.82%) | 61 (12.92%) | 5 |
| BPPV | 150 (9.77%) | 1 (0.33%) | 143 (26.83%) | 0 (0.00%) | 1 (0.21%) | 5 |
| Acute unilateral vestibulopathy (eg, vestibular neuritis) | 148 (9.64%) | 127 (41.91%) | 9 (1.69%) | 2 (2.82%) | 8 (1.69%) | 2 |
| TIA | 77 (5.02%) | 8 (2.64%) | 55 (10.32%) | 2 (2.82%) | 9 (1.91%) | 3 |
| Dysautonomia | 63 (4.10%) | 0 (0.00%) | 14 (2.63%) | 1 (1.41%) | 47 (9.96%) | 1 |
| Vestibular migraine | 35 (2.28%) | 1 (0.33%) | 31 (5.82%) | 1 (1.41%) | 1 (0.21%) | 1 |
| Menière's disease | 22 (1.43%) | 1 (0.33%) | 20 (3.75%) | 0 (0.00%) | 0 (0.00%) | 1 |
| PPPD | 22 (1.43%) | 1 (0.33%) | 2 (0.38%) | 9 (12.68%) | 7 (1.48%) | 3 |
| Tumour | 17 (1.11%) | 3 (0.99%) | 1 (0.19%) | 2 (2.82%) | 10 (2.12%) | 1 |
| Trauma | 13 (0.85%) | 0 (0.00%) | 1 (0.19%) | 0 (0.00%) | 9 (1.91%) | 3 |
| Medical side effects | 11 (0.72%) | 0 (0.00%) | 2 (0.38%) | 0 (0.00%) | 8 (1.69%) | 1 |
| Heart disease | 10 (0.65%) | 1 (0.33%) | 0 (0.00%) | 0 (0.00%) | 8 (1.69%) | 1 |
| Labyrinthitis | 9 (0.59%) | 7 (2.31%) | 1 (0.19%) | 1 (1.41%) | 0 (0.00%) | 0 |
| Infectious disease | 7 (0.46%) | 6 (1.98%) | 0 (0.00%) | 1 (1.41%) | 0 (0.00%) | 0 |
| Metabolic | 7 (0.46%) | 1 (0.33%) | 0 (0.00%) | 0 (0.00%) | 5 (1.06%) | 1 |
| Neurodegenerative disease | 5 (0.33%) | 1 (0.33%) | 0 (0.00%) | 0 (0.00%) | 4 (0.85%) | 0 |
| Acoustic neuroma | 4 (0.26%) | 1 (0.33%) | 0 (0.00%) | 2 (2.82%) | 1 (0.21%) | 0 |
| Vestibular paroxysmia | 1 (0.07%) | 0 (0.00%) | 1 (0.19%) | 0 (0.00%) | 0 (0.00%) | 0 |
| Others | 110 (7.17%) | 13 (4.29%) | 10 (1.88%) | 8 (11.27%) | 67 (14.19%) | 12 |
| Unknown | 692 (45.08%) | 62 (20.46%) | 239 (44.84%) | 44 (61.97%) | 228 (44.31%) | 119 |
| Total* | 1558 | 311 | 539 | 75 | 474 | 159 |

*Since several diagnoses can be selected per case, there are more diagnoses than cases. For each diagnosis the corresponding syndrome is listed in the table, so the total number of the syndromes is higher.

AIS, acute imbalance syndrome; AVS, acute vestibular syndrome; BPPV, benign paroxysmal positional vertigo; CVS, chronic vestibular syndrome; ED, emergency department; EVS, episodic vestibular syndrome; PPPD, persistent postural-perceptual dizziness; TIA, transient ischaemic attack.

## DISCUSSION

One-fifth to one-third of patients with dizziness presented symptoms consisting of AVS or EVS. Another one-third of patients were not classifiable based on current criteria. Patients with CVS were noticeably less likely to present to the ED. In more than one-third of the cases, which received a follow-up, the diagnosis was changed. Diagnostic uncertainty could be resolved at the follow-up visit in more than half of patients with unknown or unclear diagnoses. We found that a great number of imaging studies were ordered for dizziness work-up.

**Table 2** Cross-tabulation: vestibular syndrome ED versus follow-up (n=662)

| | | Follow-up | | | | | | |
|---|---|---|---|---|---|---|---|---|
| | | AVS | EVS | CVS | AIS | Unclear | Total | Change of syndrome (%) |
| ED | AVS | 215 | 5 | 0 | 1 | 1 | 222 | 3.15 |
| | EVS | 5 | 187 | 0 | 2 | 0 | 194 | 3.61 |
| | CVS | 0 | 0 | 34 | 0 | 0 | 34 | 0.00 |
| | AIS | 1 | 6 | 3 | 150 | 3 | 163 | 7.98 |
| | Unclear | 4 | 6 | 2 | 5 | 32 | 49 | 34.69 |
| | Total | 225 | 204 | 39 | 158 | 36 | 662 | |

AIS, acute imbalance syndrome; AVS, acute vestibular syndrome; CVS, chronic vestibular syndrome; ED, emergency department; EVS, episodic vestibular syndrome.

**Table 3** Number of diagnostic errors, change of diagnosis rates, missed dangerous diagnoses and mimics

| ED diagnoses | Total ED | No of diagnostic errors | Change of diagnosis* | No of missed strokes or TIA | Frequency of undiagnosed underlying diseases (top 3)† |
|---|---|---|---|---|---|
| Stroke/minor stroke | 132 | 18 | 13.6% | 5 (TIA) | TIA (5) Acute unilateral vestibulopathy (4) Dysautonomia (1) |
| Acute unilateral vestibulopathy (eg, vestibular neuritis) | 89 | 14 | 15.7% | 3 | Stroke/minor stroke (2) Menière's disease (2) Others (2) |
| TIA | 49 | 15 | 30.6% | 7 (strokes) | Stroke/minor stroke (7) BPPV (1) Metabolic (1) Medical side effects (1) |
| BPPV | 28 | 8 | 28.6% | 2 | Acute unilateral vestibulopathy (3) Stroke/minor stroke (2) Others (2) |
| Menière's disease | 15 | 4 | 26.7% | 0 | Acute unilateral vestibulopathy (3) Labyrinthitis (1) |
| Tumour | 14 | 1 | 7.1% | 0 | 0 |
| Vestibular migraine | 12 | 3 | 25.0% | 0 | Others (2) PPPD (1) |
| Dysautonomia | 9 | 3 | 33.3% | 0 | Others (2) Heart disease (2) Medical side effects (1) |
| Labyrinthitis | 7 | 2 | 28.6% | 0 | Acute unilateral vestibulopathy (1) Acoustic neuroma (1) |
| Infectious disease | 6 | 3 | 50.0% | 0 | Acute unilateral vestibulopathy (3) |
| Heart disease | 5 | 0 | 0.0% | 0 | 0 |
| PPPD | 5 | 0 | 0.0% | 0 | 0 |
| Others‡ | 42 | 4 | 9.5% | 0 | Dysautonomia (2) BPPV (1) Tumour (1) |
| Unknown | 249 | 145 | 58.2% | 23 | Acute unilateral vestibulopathy (35) Vestibular migraine (22) Stroke/minor stroke (14) TIA (9) |
| Total | 662 | 222 | 31.4% | 40 | |

*Since multiple answers were possible for the diagnoses, the number of diagnostic errors did not necessarily correspond to the proportion of change of diagnosis. The rate of changes of diagnoses at follow-up is calculated as follows: 100*(1−correct diagnoses/total diagnoses ED).
†Undiagnosed underlying diseases: this column shows the most frequently changed diagnosis based on the follow-up examination.
‡Diagnoses less frequent than five are not listed in the table.
BPPV, benign paroxysmal positional vertigo; ED, emergency department; PPPD, persistent postural-perceptual dizziness; TIA, transient ischaemic attack.

### Prevalence of vestibular syndromes and underlying diagnoses

The reported prevalence of AVS in the literature ranges from 10% to 22%,[2 15] which matches our findings in the ED (20%). Our reported prevalence in the ED is not generalisable to other settings such as outpatient clinics, where the proportion of CVS might predominate. Violent vertigo attacks in patients with recurrent vertigo (EVS) might prompt patients to visit the ED rather than an outpatient clinic resulting in a high prevalence of 35%. The most common ED diagnoses in the total ED population were stroke/minor stroke, BPPV and acute unilateral vestibulopathy, which are in agreement with other reports.[16 17] The posterior canal BPPV is the most common with 85%–95% of BPPV cases. It can be diagnosed with the Dix-Hallpike manoeuvre which provokes a pathognomonic torsional upbeat nystagmus.[18] If spontaneous nystagmus is present, a diagnosis other than posterior BPPV should be considered and positional testing is not advised. The ED prevalence of strokes/minor stroke was 10% in our study, which is considerably higher than previously described (~4% cerebrovascular).[16 19 20] The

**Table 4** Unknown ED diagnoses resolved after follow-up

| Diagnoses at follow-up | Unknown ED diagnoses (n=249) | Frequency |
|---|---|---|
| Acute unilateral vestibulopathy (eg, vestibular neuritis) | 35 | 14.06% |
| Others | 28 | 11.24% |
| Vestibular migraine | 22 | 8.84% |
| Stroke | 14 | 5.62% |
| TIA | 9 | 3.61% |
| Dysautonomia | 8 | 3.21% |
| Menière's disease | 8 | 3.21% |
| BPPV | 6 | 2.41% |
| PPPD | 6 | 2.41% |
| Unknown aetiology central vestibular syndrome | 4 | 1.61% |
| Metabolic disorders | 3 | 1.20% |
| Tumour | 1 | 0.40% |
| Medical side effects | 1 | 0.40% |
| Heart disease | 1 | 0.40% |
| Labyrinthitis | 1 | 0.40% |
| Infectious disease | 1 | 0.40% |
| Trauma | 0 | 0.00% |
| Neurodegenerative disease | 0 | 0.00% |
| Acoustic neuroma | 0 | 0.00% |
| Unknown | 104 | 41.77% |

BPPV, benign paroxysmal positional vertigo; ED, emergency department; PPPD, persistent postural-perceptual dizziness; TIA, transient ischaemic attack.

reported prevalence, however, is consistent with our previous, retrospective study from the same centre with another sample.[21] In patients with AVS, however, the prevalence of stroke is significantly higher at 25.4% probably due to a referral bias of a tertiary care centre including the largest stroke centre of the country. Despite extensive investigations reflected in the resources used, almost half of the cases remained undiagnosed, which is higher compared with 22% in another cross-sectional study.[16] One reason for the higher number of 'unknown' causes could be due to the applied classification rules classifying patients with multiple differential diagnoses as 'unknown'.

### Accuracy of syndrome classification

Overall, the accuracy of the classification into three different vestibular syndromes was high. In one-tenth of the cases, the documented history was not sufficient to decide whether the patient had vestibular syndrome. Possible reasons for this were a lack of documentation or an inappropriate history taking. In the group with a follow-up examination, more than one-third of the unclear ED cases could be assigned to a vestibular syndrome or a vestibular syndrome could be excluded based on the extended history of the follow-up report. This finding emphasises the importance of taking a targeted history (asking timing and triggers)[11 22] and the need of a follow-up to better assess the time course of dizziness. Digital decision support tools might assist physicians to take a structured and complete history. It is, therefore, important to improve digital competencies in the future.[23] Overall, there were only a few misclassifications of vestibular syndromes in the ED. Patients with misclassified EVS presenting initially as AVS had a short duration of symptoms which abated after the ED discharge. Diagnoses with EVS being at risk for misclassification as AVS included vestibular migraine, Menière's disease and TIA. The main reason for misclassification was the first time occurrence of episodic dizziness with no previous history of dizzy episodes as mandated by international diagnostic criteria.[24 25] We also found misclassifications of AVS as EVS in patients with cerebral strokes, vestibular neuritis and dysautonomia. Infarctions in the cerebellum (mainly PICA territory, posterior inferior cerebellar artery) can mimic positional vertigo, known as pseudo-BPPV.[26] Finally, each patient with an AVS suffers from motion intolerance, which can be misinterpreted as positional vertigo.

**Table 5** ED resources stratified by diseases (n=1535)

| | Stroke/minor stroke | BPPV | Acute unilateral vestibulopathy | TIA | Menière's disease | PPPD | Trauma |
|---|---|---|---|---|---|---|---|
| MRI | 139 (89.7%) | 62 (41.3%) | 71 (48.0%) | 62 (80.5%) | 11 (50.0%) | 9 (40.9%) | 3 (23.1%) |
| CT | 26 (16.8%) | 12 (8.0%) | 10 (6.8%) | 13 (16.9%) | 0 (0.0%) | 0 (0.0%) | 6 (46.2%) |
| Audiology | 5 (3.2%) | 16 (10.7%) | 90 (60.8%) | 6 (7.8%) | 12 (54.5%) | 1 (4.5%) | 0 (0.0%) |
| Caloric | 8 (5.2%) | 26 (17.3%) | 115 (77.7%) | 11 (14.3%) | 9 (40.9%) | 2 (9.1%) | 0 (0.0%) |
| vHIT | 4 (2.6%) | 6 (4.0%) | 41 (27.7%) | 3 (3.9%) | 2 (9.1%) | 1 (4.5%) | 0 (0.0%) |
| Total diagnoses | 155 | 150 | 148 | 77 | 22 | 22 | 13 |

BPPV, benign paroxysmal positional vertigo; ED, emergency department; PPPD, persistent postural-perceptual dizziness; TIA, transient ischaemic attack; vHIT, video head impulse test.

## Diagnostic errors in patients with dizziness

The terminology and definitions regarding diagnostic errors are under debate.[27] It can be used as an umbrella term including preventable, reducible or unavoidable diagnostic errors.[28] Our data, however, were not sufficient to assess the underlying diagnostic processes and work-ups leading to a specific diagnosis. We avoided, therefore, terms such as 'misdiagnosis', because such conclusions might be perceived as implicating errors in the diagnostic process, which we did not investigate. A subclassification into diagnostic process failure or diagnostic label failure was not possible based on our design. Diagnosing dizziness is a challenge for ED physicians and diagnostic errors are unavoidable even for experts in the field (following an optimal diagnostic process) due to the nature and complexity of the underlying diseases.[29] Thus, we aim to increase awareness about an unresolved issue regarding diagnostic accuracy in patients with dizziness visiting the ED. In a German retrospective study, 124 of 475 patients with dizziness (26%) received follow-up.[17] This number is lower than the number of patients followed up in our study (43.1%). This selection bias has to be kept in mind, interpreting the presented results. The decision to schedule patients for follow-up could reflect an intimate uncertainty with the diagnosis or be an expression of increased caution of the treating physician with that particular patient. In another study from our department on diagnostic errors, the 'feeling of atypical presentation' was the only predictor of a diagnostic error.[30] This 'feeling of atypical presentation' is likely to prompt follow-up visits leading to a selection bias in our follow-up patients. We cannot exclude any change in diagnosis within the observation period of 90 days; however, the occurrence of a second cause of dizziness unrelated to the initial diagnosis is very unlikely. In the German study, ED diagnosis was corrected in 43%.[17] We observed a lower rate of diagnostic errors in our study (31%). Of the benign ED diagnoses, 6% (n=7 of 124) were finally diagnosed with a dangerous diagnosis during follow-up in the German study[17] compared with 4% (n=29 of 662) in our study. Patients in our study, however, received significantly more often MRIs in the ED (58% MRI vs 18%). Another study reported a higher stroke misdiagnosis rate;[20] however, ED physician misdiagnosis rate was based on retrospective chart reviews derived from non-academic community hospitals with limited access to neuroimaging and neurology expertise. This might contribute to the higher number of missed dangerous diagnoses (diagnosis-related harm). Despite extensive ED work-ups in our study (including neuroimaging), four patients were still diagnosed with vestibular neuritis or BPPV and finally had a stroke (pseudo-neuritis or pseudo-BPPV) without any focal neurological signs. Recent literature confirms that 50% of patients with vestibular strokes might have isolated dizziness.[31 32] The MRI misses 10%–20% of strokes presenting with AVS during the first 24–48 hours after onset.[33] Up to 50% false-negative MRIs are reported for smaller vestibular strokes (<1 cm).[31] The 'HINTS' examination can be a possible solution to this dilemma. This three-step bedside examination, introduced in 2009,[34] includes the head impulse test, nystagmus test and test of skew and is more sensitive to stroke than early MRI. The application of a portable device using an eye-tracker and head accelerometers allows a quantitative and accurate stroke prediction in patients with AVS.[7 35–37] The comparison between diagnoses at the index (ED) and the follow-up visit shows that in many cases a definite diagnosis can only be made over time. This is often due to diagnostic criteria that require repetitive episodes of vertigo.[24 25] Some patients are symptom-free in the interval between episodes of dizziness or at the time of the emergency visit.

## ED resource use

Altogether, neuroimaging was ordered in 70% of cases, of which 83% were MRIs. This high percentage may be due to the 7/24-availability of MRI in our university hospital. We observed that a large number of MRI was performed in patients who finally received a peripheral vestibular diagnosis such as BPPV, Menière's disease or an acute unilateral vestibulopathy. The diagnosis of vestibular disorders can often be established by targeted history taking and clinical examination. There is no need for neuroimaging in clinical diagnoses such as BPPV with a typical history and typical positional nystagmus elicited by diagnostic manoeuvres.[38] Atypical findings (eg, in BPPV with apogeotropic nystagmus) or a diagnosis of exclusion (eg, in Menière's disease) might still justify neuroimaging (MRI) in the ED. CT scans, however, are only suggested in patients with suspected trauma, haemorrhage or in patients with a contraindication for an MRI. The current clinical approach leads to an unnecessary overuse of CT and MRI and increases costs exceeding billions of dollars in the US alone.[39] Patients with dizziness have longer average ED stays than patients without dizziness because they undergo more testing.[16] The rate of undiagnosed or misclassified patients remains high, resulting in higher costs and considerable waste of resources in the ED in Switzerland.[39–41] Furthermore, the overuse of CT and MRI may decrease the access for other patients and it can increase the exposition to an unnecessary amount of radiation.

## Strengths and limitations

The strengths of the study are a large number of included and screened cases and the determination of vestibular syndromes based on history and follow-up assessments. A more accurate classification of the vestibular syndromes would need, however, a prospective longitudinal study design. We also observed a referral bias (tertiary referral centre) leading to a higher proportion of dangerous diagnoses in patients with dizziness. In addition, the treating clinician decided whether a follow-up was pursued, which may have caused a selection bias.

## Implications for clinicians

Our study confirms that about one-fifth of patients suffers from AVS. The high prevalence of strokes in patients with continuous dizziness (25%) and the high number of undiagnosed or misclassified cases should increase the overall awareness regarding diagnostic errors and stroke mimics. Consequently, we suggest a three-stage diagnostic test process for patients presenting with dizziness in the ED. This approach does intend to increase diagnostic accuracy and reduce neuroimaging in the acute stage. We suggest, therefore,

(1) a more sensitive screening (triage) test including a classification into vestibular syndromes (targeted history) and recording of spontaneous nystagmus, (2) a targeted clinical examination with either 'HINTS' test[34] in patients with AVS or 'Dix-Hallpike' examination[38] in patients with EVS with triggers and (3) a dedicated neuroimaging (eg, acute and delayed MRI) in patients with suspected central causes of vertigo. Furthermore, additional tests such as the Bucket Test[42] or stance and gait tests (searching for truncal ataxia)[43] can further increase the sensitivity for the detection of patients with stroke.

In patients with EVS and absence of triggers (suspected Menière's disease or vestibular migraine), we alternatively suggest as a second-stage caloric testing and audiometry in a planned follow-up and as a third stage a delayed neuroimaging (diagnosis of exclusion). Patients without any nystagmus (spontaneous or after provocation) might need a more extended neurological examination such as BE-FAST.[44] Patients with inconclusive or atypical findings might need further assessment for risk factors (eg, ABCD2 score)[45] to minimise the risk for missed minor strokes and to prevent future harmful events. We further recommend a low threshold for organising a follow-up appointment in patients with dizziness since the symptoms and diagnosis might change over time. This study paves the way for future studies providing epidemiological data including the expected prevalence for each type of vestibular syndrome.

## CONCLUSION

One-fifth of patients with dizziness in the ED presented with AVS with a high prevalence (10%) of vestibular strokes. EVS was more frequent; however, the rate of undiagnosed patients with dizziness and the number of patients receiving neuroimaging were high. Almost half of them still remained without diagnosis and among those diagnosed were often misclassified. Many unclear cases of vertigo could be diagnostically clarified after a follow-up visit.

**Author affiliations**
[1]Department of Otorhinolaryngology, Head and Neck Surgery, Inselspital, University Hospital Bern and University of Bern, Bern, Switzerland
[2]Department of Diagnostic and Interventional Neuroradiology, Inselspital, University Hospital Bern and University of Bern, Bern, Switzerland
[3]Department of Emergency Medicine, Inselspital, University Hospital Bern and University of Bern, Bern, Switzerland
[4]Department of Neurology, Inselspital, University Hospital Bern and University of Bern, Bern, Switzerland

**Acknowledgements** We thank Dr Wolf Hautz for his valuable suggestions and review of the manuscript.

**Contributors** AK, EZ, FN and LC collected and processed the data. GM and LC conceived the study, analysed and interpreted the data and wrote the draft. MDC, TCS, FW and SJ were involved in the interpretation of the data and in the review. All authors discussed the results, commented on the manuscript and read and approved the final version. The guarantor (GM) accepts full responsibility for the finished work and/or the conduct of the study, had access to the data, and controlled the decision to publish.

**Funding** This study was supported by the Swiss National Science Foundation (No: 320030_173081).

**Competing interests** None declared.

**Patient and public involvement** Patients and/or the public were not involved in the design, conduct, reporting or dissemination plans of this research.

**Ethics approval** This study involves human participants and was approved by the local ethics committee Kantonale Ethikkommission Bern, No: 2021-00918). Given the retrospective nature of the study, informed consent was provided through a hospital-wide general consent. Patients who withdrew consent for evaluation of their medical data were excluded in accordance with legal requirements.

**Provenance and peer review** Not commissioned; externally peer reviewed.

**Data availability statement** Data are available on reasonable request. The datasets used and/or analysed during the current study are available from the corresponding author on reasonable request.

**ORCID iD**
Georgios Mantokoudis http://orcid.org/0000-0003-2268-7811

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
