## [Reviewer comments · BMJ Open]

ARTICLE DETAILS

TITLE (PROVISIONAL)	Vestibular syndromes, diagnosis and diagnostic errors in dizzy patients presenting to the emergency department. A cross-sectional study.
AUTHORS	Comolli, Lukas; Korda, Athanasia; Zamaro, Ewa; Wagner, Franca; Sauter, Thomas; Caversaccio, M; Nikles, Florence; Jung, Simon; Mantokoudis, Georgios

VERSION 1 – REVIEW

REVIEWER	Kammerlind, Ann-Sofi Linköping University, Department of Health, Medicine and Caring Sciences
REVIEW RETURNED	25-May-2022

GENERAL COMMENTS	Thank you for the opportunity to read this important, interesting and well written study. I have only a few comments: Methods, lines 100-101: When, how, why and how often did the researchers get direct information from the emergency physician? Did not these patients have dizziness as a chief complaint or a suspected diagnosis? Methods, line 107: Patients had a follow-up visit within 90 days. But there is no information about the mean time to follow-up? Results and discussion: About diagnostic errors: is it possible that some of these cases were not diagnostic errors, but a change of diagnosis from the ED visit to the follow-up? Discussion, lines 316-331: Regarding ED resource use; other effects than higher costs should be mentioned as effects of neuroimaging overuse. Patients are exposed to an unnecessary amount of radiation, and access to X-rays may decrease for other patient groups.
--

REVIEWER	Fattal, Deema The University of Iowa, Neurology , Otolaryngology
REVIEW RETURNED	31-Oct-2022

GENERAL COMMENTS	I reviewed the manuscript titled “Vestibular syndromes, diagnosis and diagnostic errors in dizzy patients presenting to the emergency department”. It is well written and addresses an ongoing difficulty in the dizziness literature, namely, how to help clinicians to minimize missing strokes/TIA in the dizzy patient in the emergency room. Minor suggestions: 1-In the abstract, if possible, to explain more on AVS, EVS, CVS diagnoses/add an example, if possible, to make those diagnoses clearer to the reader.
---

	2-I suggest the authors review/add references below. These might be useful when suggesting an algorithm on how to triage dizzy patients. ---Amir Shaban. The Bucket Test Improves Detection of Stroke in Patients With Acute Dizziness. J Emerg Med. 2021 Apr;60(4):485-494.doi: 10.1016/j.jemermed.2020.10.052. Epub 2020 Dec 8. ---Kerber, Kevin A., et al. "Stroke risk stratification in acute dizziness presentations: a prospective imaging-based study." Neurology 85.21 (2015): 1869-1878. ---Carmona, Sergio, et al. "The diagnostic accuracy of truncal ataxia and HINTS as cardinal signs for acute vestibular syndrome." Frontiers in neurology 7 (2016): 125. ---Kerber et al. Stroke among patients with dizziness, vertigo, and imbalance in the emergency department: a population-based study. Stroke 2006 Oct;37(10):2484-7. doi: 10.1161/01.STR.0000240329.48263.0d. Epub 2006 Aug 31. 3-I suggest that the authors elaborate on how dysautonomia was diagnosed. 4-I suggest that the authors further emphasize how posterior canal BPPV, which is 90% of BPPV, has a straightforward diagnosis-by performing the Dix Hallpike- and how it has a pathognomonic nystagmus (torsional to floor/upbeat to forehead) to add to educational value of the article. Furthermore, I recommend mentioning that if someone has spontaneous nystagmus then that patient should not have Dix Hallpike performed. 5- 29/662 4.4% missed TIA/Strokes; this is much lower than Kerber's findings (2006) of 35%; any reason for that? Thanks
--	--

VERSION 1 – AUTHOR RESPONSE

Reviewer 1 (Dr. Ann-Sofi Kammerlind, Linköping University)

1. Methods, lines 100-101: When, how, why and how often did the researchers get direct information from the emergency physician? Did not these patients have dizziness as a chief complaint or a suspected diagnosis?

Response:

We thank the reviewer for this insightful comment. A research fellow screened the ED clinical information system for chief complaints including keywords such as “dizziness”, “vertigo”, “unsteadiness”, “presyncope”, “vomiting”, “nausea”. Patients fulfilling these criteria were screened and clinically assessed in person in order to identify eligible subjects for the study. ED physicians

reported patients with the same screening criteria (chief complaint dizziness) as well. We added these details in the methods section.

2. Methods, line 107: Patients had a follow-up visit within 90 days. But there is no information about the mean time to follow-up?

Response:

Thank you for this question. We coded the time point of the follow-up exam as a binary variable (90d follow-up yes/no) and could therefore not calculate the mean follow-up interval. We would be happy to calculate the mean time to follow-up, however, we would need an extension of the manuscript revision deadline in order to add the absolute follow-up dates in our database for all 662 patients. Please advice, whether this additional information still needs to be added in the manuscript.

3. Results and discussion: About diagnostic errors: is it possible that some of these cases were not diagnostic errors, but a change of diagnosis from the ED visit to the follow-up?

Response:

Thank you for pointing this out. We cannot absolutely exclude any possible change in diagnosis from the ED visit to the follow-up, however, the occurrence of a second cause of dizziness unrelated to the initial diagnosis is very unlikely. We, therefore, decided that a change of diagnosis in the follow up indicates that the initial diagnosis was wrong and thus a diagnostic error. Our study design does not allow us to differentiate the type of diagnostic error. But the rate of diagnostic errors indicates the difficulty of diagnosing a dizziness patient in the emergency department. For better understanding we added: "*If the diagnosis changed in the follow-up we defined it as a diagnostic error.*" (p. 6, line 168) and mention this issue in the discussion.

4. Discussion, lines 316-331: Regarding ED resource use; other effects than higher costs should be mentioned as effects of neuroimaging overuse. Patients are exposed to an unnecessary amount of radiation, and access to X-rays may decrease for other patient groups.

Response:

We thank you for the valuable additions. We added the discussion points on p. 14 lines 346-348.

“Furthermore, the overuse of computed tomography and magnetic resonance imaging may decrease access for other patients and it can increase the exposition to an unnecessary amount of radiation.”

Reviewer 2 (Dr. Deema Fattal, The University of Iowa, Iowa City VA Medical Center)

1. In the abstract, if possible, to explain more on AVS, EVS, CVS diagnoses/add an example, if possible, to make those diagnoses clearer to the reader.

Response:

Thank you for your insightful comment. We agree with you, and we added at least some diagnoses for each syndrome. Since the word count for the abstract is limited only a small change was possible.

"Vestibular syndromes are differentiated in acute (e.g stroke, neuritis vestibularis), episodic (e.g. BPPV, TIA) and chronic (e.g. PPPD) vestibular syndrome." (p.2, links 36-38)

2. I suggest the authors review/add references below. These might be useful when suggesting an algorithm on how to triage dizzy patients.

a) ---Amir Shaban. The Bucket Test Improves Detection of Stroke in Patients With Acute Dizziness. J Emerg Med. 2021 Apr;60(4):485-494.doi: 10.1016/j.jemermed.2020.10.052. Epub 2020 Dec 8.

b) ---Kerber, Kevin A., et al. "Stroke risk stratification in acute dizziness presentations: a prospective imaging-based study." Neurology 85.21 (2015): 1869-1878.

c) ---Carmona, Sergio, et al. "The diagnostic accuracy of truncal ataxia and HINTS as cardinal signs for acute vestibular syndrome." Frontiers in neurology 7 (2016): 125.

d) ---Kerber et al. Stroke among patients with dizziness, vertigo, and imbalance in the emergency department: a population-based study. Stroke 2006 Oct;37(10):2484-7. doi: 10.1161/01.STR.0000240329.48263.0d. Epub 2006 Aug 31.

Response:

We thank you for the valuable references. We added the references as suggested by the reviewer.

3. I suggest that the authors elaborate on how dysautonomia was diagnosed.

Response:

We thank you for your insightful comment. We added the following text. "*A dysautonomia was diagnosed when the "Schellong test" was positive.*" (p.7, lines 184-185)

4. I suggest that the authors further emphasize how posterior canal BPPV, which is 90% of BPPV, has a straightforward diagnosis-by performing the Dix Hallpike- and how it has a pathognomonic nystagmus (torsional to floor/upbeat to forehead) to add to educational value of the article.

Furthermore, I recommend mentioning that if someone has spontaneous nystagmus then that patient should not have Dix Hallpike performed.

Response:

We thank you for your insightful comment. We added the following text to the discussion.

“The posterior canal BPPV is the most common with 85-95% of BPPV cases. It can be diagnosed with the Dix-Hallpike maneuver which provokes a pathognomonic torsional upbeat nystagmus. If spontaneous nystagmus is present, a diagnosis other than posterior canal BPPV should be considered and positional testing is not advised.” (p. 12, lines 257-260)

5. 29/662 4.4% missed TIA/-strokes; this is much lower than Kerber's findings (2006) of 35%; any reason for that?

Response:

Thank you for your insightful comment. The data in Kerber's study was collected from 2000-2003 using data from non-academic community hospitals from chart records provided by ED physicians without an onsite neurology support. The majority of cases did not receive brain imaging and misdiagnosis rate was derived by the review of the medical records. Our hospital is a referral stroke center with a 24/7 MRI availability and onsite neurology coverage. Therefore, the lower rate of stroke misdiagnosis seems plausible. We added this in the discussion.

VERSION 2 – REVIEW

REVIEWER	Kammerlind, Ann-Sofi Linköping University, Department of Health, Medicine and Caring Sciences
REVIEW RETURNED	03-Feb-2023
GENERAL COMMENTS	I am happy with the revisions made.